# Small-Molecules as Chemiluminescent Probes to Detect Lipase Activity

**DOI:** 10.3390/ijms23169039

**Published:** 2022-08-12

**Authors:** Paolo La Rocca, Alessandra Mingione, Silvana Casati, Roberta Ottria, Pietro Allevi, Pierangela Ciuffreda, Paola Rota

**Affiliations:** 1Dipartimento di Scienze Biomediche per la Salute, Università degli Studi di Milano, 20133 Milano, Italy; 2Dipartimento di Scienze Biomediche e Cliniche, Università di Milano, 20157 Milano, Italy; 3Dipartimento di Scienze Biomediche, Chirurgiche ed Odontoiatriche, Università degli Studi di Milano, 20133 Milano, Italy

**Keywords:** chemiluminescence, 1,2-dioxetanes, lipase, glow-luminescence, flash-luminescence

## Abstract

The set-up of highly sensitive detection tools to evaluate lipase activity remains a central goal in different fields. In this context, we proposed new chemiluminescent 1,2-dioxetane luminophores, sharing an octanoyl triggerable group, to monitor lipase activity. We herein report the synthesis and both the evaluation of their luminescence emission profile and their enzyme–substrate specificity, generated by three different commercial lipases (*Candida cylindracea*, *Pseudomonas fluorescens*, and *Mucor miehei*) and one esterase (porcine liver esterase, PLE, as a literature control). Remarkably, the present study confirmed the applicability of these 1,2-dioxetane luminophores as (i) highly efficient, broad-range, chemiluminescent probes for the detection and the enzymatic activity evaluation of lipases and as (ii) promising candidates for the future development of both flash- and glow-type luminescence assays.

## 1. Introduction

Biosensing and bioimaging in the industrial, environmental, and medical fields need a constant supply of more sensitive diagnostic tools [1,2,3,4,5,6,7,8]. In this regard, luminescence-based techniques (i.e., chemiluminescence) are usually preferred over fluorescence-based ones because they do not require any irradiation from external sources, increasing the sensitivity [7,8]. Recently, chemiluminescence, which is a light emission associated with chemiexcitation due to a chemical reaction yielding an electronically excited product, has attracted a lot of interest [9,10,11,12,13,14,15,16,17,18,19,20]. Chemiluminescence applications to biochemical and pharmacological or clinical research and diagnostics, indeed, have increased in the last decade [21].

Industrial biotechnologies could take advantage of these strategies in the set-up of faster and simpler protocols for enzymatic activity evaluation. In this context, lipases, one of the most versatile class of enzymes available in nature, play a pivotal role in industrial production [22,23,24,25]. Considering the growing exploitation of these enzymes in new industrial processes, the set-up of protocols based on innovative and more specific substrates for their detection is of great importance.

A significant class of chemiluminescent molecules, useful as enzymatic substrates, is represented by the adamantylidene 1,2-dioxetane luminophores (Figure 1), firstly discovered by Schaap et al. [26,27,28]. The chemiexcitation process of these substrates is due to the formation of high energy phenolate ion generated after the cleavage of a triggerable group by a specific enzyme (Figure 1A). 

The direct chemiluminescence production, obtained after the cleavage of the trigger, enables the set-up of simple methods for enzymatic activity evaluation. This issue allows us, indeed, to overcome the use of multi enzymatic step reactions, as in the case of luciferin derivative probes [7,29], or secondary chemical reactions, such as oxidation in the case of luminol derivatives, as the base of numerous currently used methods [21].

However, while the first two generations [26,27,28] of phenoxy-1,2-dioxetane luminophores showed weak chemiluminescence emission under aqueous conditions, this problem has been overcome in the new, recently proposed derivatives [8,30], making them suitable for bioimaging in vitro in cell cultures and in vivo in animal models.

In the present study, continuing our interest in lipase selective probe development [31,32,33,34,35] suitable for in vitro and in vivo assays, we investigated a group of phenoxy-1,2-dioxetane luminophores, **2**–**4**, carrying an octanoyl chain as a triggerable substrate. The enzymatic cleavage of these phenoxy-dioxetane luminophores generates a direct chemiluminescence emission which is strongly dependent on the lipase activity. Therefore, starting from compound **1**, we synthesized and fully characterized two unreported probes, compounds **2** and **3** (Figure 1B), together with the recently patented [18,19] but not commercially available compound **4**. This molecule has been recently reported as an alternative to 4-methylumbelliferone fluorogenic probes [36,37,38] for the direct detection of pathogenic bacteria, as a substrate of Salmonella esterase [18]. Remarkably, compounds **2** and **3** presented a simplified synthesis more accessible than that performed to obtain probe **4**.

Finally, the evaluation of their luminescence emission profile in aqueous solution and enzyme–substrate selectivity, triggered by three different commercially available lipases (from *Candida cylindracea*, *Pseudomonas fluorescens*, and *Mucor miehei*) and one esterase (porcine liver esterase, as a literature control [18]), are herein reported.

## 2. Results and Discussion

### 2.1. Synthesis of Phenoxy-1,2-Dioxetane Luminophores **2**, **3**, and **4**

In this study, our interest was directed to synthesize and characterize new chemiluminescent 1,2-dioxetane probes to monitor lipase activity. Specifically, we planned a synthetic way to achieve three probes carrying an octanoyl chain, as a substrate mimic, and an *ortho* chlorine atom in the phenolic ring. 

The choice of this acyl chain derives from our previous studies [31,32,33,34,35], which have suggested medium-short chains as a versatile substrate of lipase and esterase activity evaluation, and it is also supported by the recent literature [18]. Indeed, in a rational design of a substrate for lipolytic activity evaluation, the alkyl chain length should be carefully evaluated. The diverse alkyl esters have different lipophilicity, and this could influence the enzyme interfacial activation, a complex set of chemical–physical, substrate–lipase interactions essential for their activity. Furthermore, the presence of an *ortho* chlorine atom in the aromatic ring (i) decreases the p*K_a_* of the phenolate intermediate, facilitating the activation of the chemiexcitation mechanism at a physiological pH, and (ii) increases the photostability of the luminophore [39]. 

According to these data and starting from the commercially available compound **1**, we planned the synthesis of the derivative **5** (Figure 1), the stable precursor of the unreported phenoxy-1,2-dioxetane luminophore **2**. For this purpose, compound **1** was reacted with octanoyl chloride **6** in basic solution (Et_3_N in CH_2_Cl_2_), giving the unreported intermediate **5** in good yields (79%). 

Moreover, we also planned the synthesis of compound **10** (Figure 2), the precursor of probe **3**, designed with a small spacer between the fatty acid moiety and the hydroxyl group to optimize the enzymatic recognition and the subsequent cleavage. Indeed, the presence of this spacer avoids the steric interference caused by the *ortho* substituent in the proximity of the enzyme catalytic site [40]. 

To achieve the desired compound **10**, we hypothesized the final coupling between the precursor **1** with the spaced trigger molecule **7** (Figure 2B). The synthesis of the intermediate **7** (Figure 2A) was accomplished by reacting 4-hydroxy benzyl alcohol **8** with octanoyl chloride **6** in basic medium, obtaining compound **9**. The esterification was performed in two different solvents (CH_2_Cl_2_ and THF); however, although the compound **8** is more soluble in THF, the final reaction yields remain comparable (50%) in the two solvents.

The successive conversion of compound **9** in the iodine derivative **7** was obtained, as already reported [18], but in higher yields (88% compared to 64% from the literature [18]) by treating the precursor with sodium iodide in presence of trimethylsilyl chloride. Compound **7** presents chemical–physical properties superimposable to those reported in the literature. 

Then, compound **10** was obtained in suitable yields (36%), as shown in Figure 2B, reacting the commercially available precursor **1** with the iodine derivative **7** in the presence of potassium carbonate in DMF overnight.

Finally, we also synthesized compound **11** (Figure 3), the precursor of the already published [18] but non commercially available dioxetane luminophore **4**. This molecule carries a methyl acrylate substituent at the *ortho* position of the phenoxy-dioxetane; in fact, according to recent literature, the introduction of an electron-withdrawing acrylic substituent in that position is essential to improve the quantic yield of the luminophore in aqueous solution [8,40]. Briefly, starting from compound **1** dissolved in toluene, *N*-iodosuccinimide was added at −50 °C and the temperature was gradually increased to −30 °C over a period of 3 h. Differently from the study in [40], the reaction was conducted at a lower temperature and was strictly monitored by TLC to better control the formation of the *ortho, para*-diiodo derivative, a by-product which causes a marked yield reduction. Under this reaction conditions and after chromatography purification, compound **12** was achieved in a 35% yield. 

Subsequently, intermediate **13** was reached after the reaction of derivative **12** with methyl acrylate in basic solution and in the presence of palladium acetate (Pd(OAc)_2_) and tri(*o*-tolyl)phosphine (P(*o*-tolyl)_3_) and the successive ester deprotection with sodium hydroxide. Then, this intermediate was reacted with the spacer-trigger molecule **7**, in the presence of sodium hydride, to obtain compound **11**.

As a final part of the synthetic route, all three dioxetane luminophores **2**–**4** were achieved by radical oxidation of the respective precursors **5**, **10**, and **11** in the presence of O_2_ (air bubbling), light, and hematoporphyrin as generator of singlet oxygen, as shown in Table 1. It should be noted that all final compounds **2**–**4** were obtained in good yields (52–74%) and in short reaction times. However, compound **11** was obtained with lower yields than those previously described [18], even using methylene blue as singlet molecular oxygen generator.

### 2.2. Chemiluminescence Emission Profile of **2**, **3** and **4** over Time

The chemiluminescent quantitation methods that evaluate the output in terms of luminescence quantum yield and enzyme–substrate reaction type provide highly sensitive alternatives to colorimetric and fluorescent substrates [6,7,18,36,37,38]. Luminescent labels are desirable as they provide high sensitivities due to their relatively high luminescent yields and the low naturally occurring luminescence backgrounds in biological systems [41,42].

Here we present the development of chemiluminescent detection protocols performed with the three synthesized probes in the presence of three different lipases, *Candida cylindracea* (or *Candida rugosa*, CCL), *Pseudomonas fluorescence* (PFL), *Mucor miehei* (MML), and the porcine liver esterase (PLE) as control [18]. Remarkably, we selected PLE as a model because, in the literature [18], the chemiluminescent emission profile of probe **4** has been tested only on this enzyme. It is noteworthy that no tests have been previously reported on lipases using this probe.

We chose three lipases of microbial origins [25] because they present advantages in comparison to those derived from plants or animals in terms of catalytic activity variety, high yield production, regular supply due to the absence of seasonal fluctuations, and with a very high growth rate of microorganisms in economical media [43,44]. Indeed, CCL, PFL, and MML are extensively utilized in several industrial applications and production, such as dairy, food and beverage, animal feed, cleaning, biofuel, pharmaceuticals, textiles, cosmetics, the flavor industry, biocatalytic resolution, esters and amino acid derivatives, or fine chemicals, agrochemicals, and biosensors [22,23,24].

Considering the large use of lipases in industrial production and biochemical or pharmacological research, the need for new and highly sensitive methods to detect or study their activity became clear. With this intent we selected the adamantyl-l,2-dioxetane as luminophore scaffold to synthesize the new lipophilic probes. Indeed, the hydrolytic cleavage of adamantyl-l,2-dioxetane substrates **2**–**4** by lipases results in the formation of an electron-rich dioxetane phenolate anion, which initiates a decomposition mechanism called chemically initiated electron exchange luminescence (Figure 1) [45,46]. 

The chemiluminescence emission profile of compound **2**, **3,** and **4** (final well concentration 10 μM) obtained by a time-course analysis was measured in Tris-HCl buffer (pH 7.4, 1mM EDTA, 10% DMSO) in the presence or in the absence (no enzyme) of CCL, PFL, and MML, and PLE as control (Figure 2).

The chemiluminescent signal obtained from the lipase-catalyzed dioxetane decomposition reaction for compounds **2** and **3**, upon activation with all lipases and esterase, was less intense but more stable than that of compound **4**, as shown in Figure 2a–c. This light emission signal is typically a steady-state glow that can be measured over a long time (glow luminescence). The relative light unit (RLU) can be measured for a longer time than compound **4**, which emitted brighter light quickly (flash luminescence) (Figure 2c). In the presence of substrates **2** and **3**, the chemiluminescent signal rises as the metastable dioxetane anion, produced by enzyme hydrolysis, accumulates because its production rate exceeds its decay to the ground state. When the rate of anion production equals the rate of its decay, a steady-state plateau is reached producing a prolonged glow emission that simplifies the measurement of the light signal.

These differences in light emission profiles are of fundamental importance in having tools specific for different applications. Compound **4**, thanks to its flash luminescence profile, is suitable for high throughput applications, typically on the order of seconds, requiring specific luminometers equipped with an autoinjector. On the contrary, compounds **2** and **3**, characterized by a glow luminescence profile, are more indicated for enzyme kinetic studies typical of biochemical or pharmacological research. This difference in the light emission profile observed in the new substrates **2** and **3** compared to **4** is observable both in the lipases and the esterase used as a control.

Moreover, the two newly synthesized luminophores, even if with lower RLUs displayed, are of large applicability because they do not require a specific luminometer but can be used with simpler multiplate readers. Furthermore, an important advantage of compounds **2** and **3** was that they exhibited a much lower basal luminescence background in the absence of enzymes than compound **4** (see Supplementary Materials, Appendix A).

## 3. Materials and Methods

### 3.1. Chemistry

General Information

Compound **1**, or 3-(((1*r*,3*r*,5*R*,7*S*)-adamantan-2-ylidene)(methoxy)methyl)-2-chlorophenol, was purchased from AChemBlock (Advanced ChemBlocks Inc 849 Mitten Rd, Burlingame, CA 94010, USA). All other chemicals and solvents used were of synthetic or analytical grade and purchased from Sigma-Aldrich (St. Louis, MO, USA). Deionized water was prepared by filtering water in a Milli-Q Simplicity 185 filtration system from Millipore (Bedford, MA, USA). Solvents were dried using standard methods and distilled before use. The progress of all reactions was monitored by thin-layer chromatography (TLC) carried out on 0.25 mm Sigma-Aldrich silica gel plates (60 F254) using UV light, anisaldehyde/H_2_SO_4_/EtOH solution or phosphomolybdic acid solution 10% in ethanol and heat as the developing agent. Flash chromatography was performed with normal phase silica gel (Sigma-Aldrich 230–400 mesh silica gel).

Nuclear magnetic resonance spectra were recorded at 298K on a Bruker AM-500 spectrometer equipped with a 5 mm inverse-geometry broadband probe and operating at 500.13 MHz for ^1^H and 125.76 MHz for ^13^C. Chemical shifts are reported in parts per million and are referenced for ^1^H spectra to a solvent residue proton signal (*δ* = 7.26 ppm for CDCl_3;_
*δ* = 3.31 ppm for CD_3_OD) and for ^13^C spectra to solvent carbon signal (central line at *δ* = 77.00, for CDCl_3;_
*δ* = 49.05 ppm for CD_3_OD). The ^1^H and ^13^C resonances were assigned by ^1^H-^1^H (COSY) and ^1^H-^13^C (HSQC and HMBC) correlation 2D experiments. The ^1^H NMR data are tabulated in the following order: multiplicity (s = singlet, d = doublet, t = triplet, br = broad, m = multiplet, app = apparent), and coupling constant(s) are given in Hz, number of protons, and assignment of proton(s). Mass spectrometry spectra were obtained on an ABSciex 4000Qtrap mass spectrometer equipped with an ESI ion source. The spectra were collected in continuous flow mode by connecting the infusion pump directly to the ESI source. Solutions of the compounds were infused at a flow rate of 0.01 mL min^−1^, the spray voltage was set at 5.5 kV in the positive ion mode with a capillary temperature of 550 °C. Full-scan mass spectra were recorded by scanning an *m/z* range of 100–2000.

For the obtained intermediates **5**, **10**, **11**, and the final compounds **2**–**4,** the ^1^H and ^13^C NMR assignments refer to this general structure:



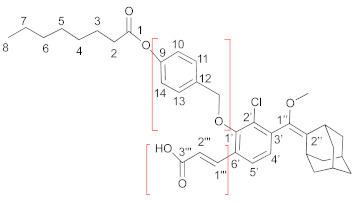



Compound **5**

Compound **1,** or 3-(((1*r*,3*r*,5*R*,7*S*)-adamantan-2-ylidene)(methoxy)methyl)-2-chlorophenol (25 mg, 0.082 mmol), was dissolved in 0.5 mL of CH_2_Cl_2_ containing Et_3_N (20 μL, 0.147 mmol). The reaction mixture was cooled to 0 ℃ and octanoyl chloride (17 μL, 0.098 mmol) was added. The solution was stirred at room temperature for 1 h, with monitoring of the reaction by TLC. Upon completion, the reaction was quenched by water (27 μL) and the solution was concentrated under reduced pressure. Then the residue was diluted with EtOAc (2.5 mL) and washed with HCl 1M (1 × 2.5 mL) and then with brine (2 × 2.5 mL). The organic phase was dried over Na_2_SO_4_ and concentrated under reduced pressure. Purification by column chromatography (hexane:EtOAc, 97:3, *v*:*v*) afforded compound **5** (28 mg, 79% yield) as colorless oil. ^1^H NMR (CDCl_3_): *δ* 7.26 (t app, *J* = 7.8 Hz, 1H, H-5′), 7.16 (dd, *J* = 7.6, *J* = 1.6 Hz, 1H, H-4′ or H-6′), 7.10 (dd, *J* = 8.0, *J* = 1.6 Hz, 1H, H-4′ or H-6′), 3.31 (s, 3H, OCH_3_), 3.28 (br s, 1H, adamantane), 2.62 (t, *J* = 7.5 Hz, 2H, H-2), 2.08 (br s, 1H, adamantane), 1.98–1.64 (overlapping, 14H, 12 adamantane and H-3), 1.48–1.24 (overlapping, 8H, octanoyl), 0.89 ppm (t, *J* = 6.8 Hz, 3H, H-8); ^13^C NMR (CDCl_3_): *δ* 171.3 (C-1), 147.4 (C-1′), 139.4 (C-1″), 136.3 (C-3′), 132.0 (C-2″), 129.3(C-4′ or C-6′), 127.6 (C-2′), 126.7 (C-5′), 123.0 (C-4′ or C-6′), 57.1 (O*C*H_3_), 39.2, 39.1, 38.6 37.1 (4C, adamantane), 34.1 (C2), 32.8 (adamantane), 31.6 (octanoyl), 29.7 (2C, adamantane, and octanoyl), 29.1, 28.9 (2C, octanoyl), 28.4, 28.2 (2C, adamantane), 24.9 (2C, adamantane, and octanoyl), 22.6 (C-7), 14.0 ppm (C-8). MS (ESI, positive) *m/z* calc. for C_26_H_35_ClO_3_: 430.2 (100%), 431.2 (28%), 432.2 (32%); found: 453.3 (100%) [M + Na]^+^, 454.2 (40%) [M + Na]^+^, 455.2 (50%) [M + Na]^+.^, respectively.

Compound **9**

4-Hydroxybenzyl alcohol, compound **8** (200 mg, 1.6 mmol), was dissolved in 5 mL of THF or CH_2_Cl_2_, and then Et_3_N (357 μL, 2.6 mmol) was added. Reaction mixture was cooled to 0 ℃ and octanoyl chloride (303 μL, 1.8 mmol) was added. The solution was stirred at room temperature for 1.5 h, with monitoring by TLC. Upon completion and quenching by water (200 μL), the solution was concentrated under reduced pressure and then diluted with CH_2_Cl_2_ (25 mL) and washed with NaHCO_3_ (25 mL) and then with brine (25 mL). The organic phase was dried over Na_2_SO_4_ and concentrated under reduced pressure. Purification by column chromatography (hexane:EtOAc, 95:5, *v*:*v*) afforded compound **9** (200 mg, 50% yield) as colorless oil. All its chemical–physical properties were superimposable with those reported in the literature [18]. ^1^H NMR (MeOD): *δ* 7.35 (overlapping, 2H, H-10, and H-14), 7.10 (overlapping, 2H, H-11, and H-13), 4.60 (s, 2H, OCH_2_), 2.62 (t, *J* = 7.5 Hz, 2H, octanoyl), 1.82 (m, 2H, octanoyl), 1.54–1.34 (overlapping, 8H, octanoyl), 0.99 ppm (t, 2H, *J* = 6.7 Hz, H-8).

Compound **7**

Starting from compound **9** (200 mg, 0.80 mmol), the compound **7** was obtained after 0.5 h, following the procedure described in the literature [18]. Briefly, compound **9** was dissolved in 4 mL of CH_3_CN and cooled to 0 °C. Then, sodium iodide (360 mg, 2.40 mmol) was added, followed by the rapid addition of TMS-Cl (306 μL, 2.4 mmol). The reaction was allowed to warm up to room temperature and monitored by TLC upon completion (0.5 h). The reaction mixture was diluted with EtOAc (25 mL) and washed with saturated Na_2_S_2_O_3_ (25 mL) followed by brine (25 mL). The organic layer was separated, dried over Na_2_SO_4_, filtered, and the solvent was evaporated under reduced pressure to afford compound **7** (250 mg, 88% yield) as white solid. All its chemical–physical properties were superimposable with those previously reported [18]. 

Compound **10**

Compound **1** (20 mg, 0.065 mmol) was dissolved in 0.3 mL of DMF and K_2_CO_3_ (18 mg, 0.13 mmol) was added. The reaction mixture was cooled to 0 ℃ and the iodo derivative **7** (26 mg, 0.07 mmol) was added. The solution was stirred at room temperature overnight, with monitoring by TLC. Upon completion, the solution was diluted with EtOAc (2.5 mL) and washed with NaHCO_3_ (2.5 mL) and then with brine (2 × 2.5 mL). The organic phase was dried over Na_2_SO_4_ and concentrated under reduced pressure. Purification was performed by column chromatography (ciclohexane:EtOAc, 95:5, *v*:*v*), followed by a second one (hexane:CH_2_Cl_2_:EtOAc, 80:15:2, *v*:*v*:*v*), affording the compound **10** (13 mg, 36% yield) as colorless oil. ^1^H NMR (CDCl_3_): *δ* 7.50 (d, *J* = 8.4 Hz, 2H, H-10, and H-14), 7.17 (t app, *J* = 7.9 Hz, 1H, H-5′), 7.10 (d, *J* = 8.4 Hz, 2H, H-11, and H-13), 6.95 (d, *J* = 8.2 Hz, 1H, H-6′), 6.89 (d, *J* = 7.5 Hz , 1H, H-4′), 5.14 (s, 2H, CH_2_O), 3.32 (s, 3H, OCH_3_), 3.28 (br s, 1H, adamantane), 2.56 (t, *J* = 7.5 Hz, 2H, H-2), 2.08 (br s, 1H, adamantane), 1.99–1.60 (overlapping, 14H, 12 adamantane, and H-3), 1.46–1.22 (overlapping, 8H, H-4, H-5, H-6, and H-7), 0.90 ppm (t, *J* = 6.4 Hz, 3H, H-8). ^13^C NMR (CDCl_3_): *δ* 172.3 (C-1), 154.3 (C-1′), 150.5 (C-9), 139.9 (C-1″), 136.2 (C-3′), 134.0 (C-12), 131.2 (C-2″), 128.2 (2C, C-10, and C-14), 126.6 (C-5′), 124.5 (C-4′), 121.7 (2C, C-11 and C-13), 115.4 (C-2′), 113.1 (C-6′), 70.4 (CH_2_O), 57.0 (OCH_3_), 39.2, 39.1, 38.7, 38.6, 37.2 (5C, adamantane, and octanoyl), 34.4 (C-2), 32.8, 31.7, 29.6, 29.1, 28.9, 28.5, 28.3 (7C, adamantane, and octanoyl), 24.9 (C-3), 22.6 (C-7), 14.1 ppm (C-8). MS (ESI positive) *m/z* calc. for C_33_H_41_ClO_4_: 536.3 (100%), 537.3 (36%), 538.3 (32%); found: 537.2 (100%) [M + H]^+^, 538.2 (35%) [M + H]^+^, 539.3 (35%) [M + H]^+^, respectively.

Compound **12**

Compound **1** (100 mg, 0.33 mmol) was dissolved in 5 mL of anhydrous toluene, cooled at −50 °C, and *N*-iodosuccinimide (41 mg, 0.18 mmol) was added in one portion. The reaction was stirred for 3 h, gradually increasing the temperature to −30 °C (to better control the diiodo by-product formation). The reaction was strictly monitored by TLC to minimize the diiodo byproduct formation. Then, Na_2_S_2_O_3_ was added to the solution, maintained by stirring at −30 °C for 10 min. It was then diluted with EtOAc (20 mL) and washed with brine (1 × 20 mL). The organic phase was dried over Na_2_SO_4_ and concentrated under reduced pressure. Purification by column chromatography (hexane:EtOAc, 10:0.3, *v*:*v*) afforded compound **14** (50 mg, 35% yield) as a white solid. ^1^H NMR (CDCl_3_): *δ* 7.61 (d, *J* = 8.1 Hz, 1H, H-5′), 6.62 (d, *J* = 8.1 Hz, 1H, H-4′), 6.13 (s, 1H, OH), 3.31 (s, 1H, OCH_3_), 3.26 (br s, 1H, adamantane), 2.09 (br s, 1H, adamantane), 2.00–1.61 ppm (overlapping, 12H, adamantane). All its chemical–physical properties were superimposable with those previously reported in the literature [40].

Compound **13**

Starting from compound **12** (50 mg, 0.12 mmol), adopting the two-step procedure described in the literature [40], compound **13** was obtained (29 mg, 67% overall yield). ^1^H NMR (CDCl_3_): *δ* 7.98 (d, *J* = 16.1 Hz, 1H, H-2″′), 7.48 (d, *J* = 8.0 Hz, 1H, H-5′), 6.83 (d, *J* = 8.0 Hz, 1H, H-4′), 6.59 (d, *J* = 16.1 Hz, 1H, H-1″′), 3.29 (s, 1H, OCH_3_), 3.26 (br s, 1H, adamantane), 2.09 (br s, 1H, adamantane), 2.01–1.67 ppm (overlapping, 12H, adamantane). All its chemical–physical properties were superimposable with those previously reported [18]. 

Compound **11**

Starting from compound **13** (25 mg, 0.07 mmol), compound **11** was obtained (11 mg, 25% yield) meticulously following the procedure described in the literature [18]: ^1^H NMR (CDCl_3_): *δ* 8.00 (d, *J* = 16.1 Hz, 1H, H-1″′), 7.49 (d, *J* = 8.4 Hz, 2H, H-10 and H-14, 7.46 (d, *J* = 8.0 Hz, 1H, H-5′), 7.11–7.07 (overlapping, 3H, H-11, H-13, and H-4′), 6.45 (d, *J* = 16.1 Hz, 1H, H-2″′), 5.02 (br s, 2H, CH_2_O), 3.34 (s, 3H, OCH_3_), 3.29 (br s, 1H, adamantane), 2.55 (t, *J* = 7.5 Hz, 2H, H-2), 2.08 (br s, 1H, adamantane), 2.01–1.23 (overlapping, 22H, 12 adamantane, and 10 octanoyl), 0.89 (tr, *J* = 6.7 Hz, 3H, H-8). All the other chemical–physical properties were superimposable with those previously reported [18].

General Procedure for the Radical Oxidation of Compounds **5**, **10**, and **11** to Obtain Final Compound **2**–**4**.

Enol ethers **5**, **10**, or **11** (0.034 mmol) and hematoporphyrin (20% *w*/*w*) were dissolved in THF. Air was bubbled through the solution while irradiating with yellow light (maintaining the reaction in ice to avoid temperature increase). Reaction was monitored by TLC until the completion. Then, the solvent was concentrated under reduced pressure to afford, after chromatography purification, the desired final compounds **2**, **3**, or **4**.

Compound **2**

Enol ether **5** (14 mg, 0.034 mmol) and a few milligrams of hematoporphyrin (3 mg, 20% *w*/*w*) were dissolved in 1 mL of THF, reacted for 1.5 h, and worked up according to the general procedure. The crude material was purified by column chromatography (hexane:EtOAc, 97:4, *v*:*v*), affording compound **2** (10 mg, 67% yield) as colorless oil. The product was isolated as a mixture of diastereomers. ^1^H NMR (CDCl_3_): *δ* 8.00 (br d, *J* = 7.7 Hz, 1H, H-4′), 7.42 (t app, *J* = 8.0 Hz, 1H, H-5′), 7.20 (dd, *J* = 8.0, *J* = 1.5 Hz, 1H, H-6′), 3.21 (s, 3H, OCH_3_), 3.01 (br s, 1H, adamantane), 2.62 (t, *J* = 7.5 Hz, 2H, H-2), 2.27 (d, *J* = 12.4 Hz, 1H, adamantane), 2.01 (br s, 1H, adamantane), 1.90–1.24 (overlapping, 21H, adamantane, and octanoyl), 0.89 ppm (t, *J* = 6.8 Hz, 3H, H-8); ^13^C NMR (100 MHz, CDCl_3_): *δ* 171.3 (C-1), 148.3 (C-1′), 133.6 (C-3′), 130.7 (C-4′), 127.2 (C-5′), 125.5 (C-2′), 125.1 (C-6′), 111.8 (C-1″), 96.2 (C-2″), 49.7 (OCH_3_), 36.6, 34.0, 33.9 (3C, adamantane, and octanoyl), 33.5 (C-2), 32.3, 32.3, 31.6, 31.6, 31.5, 29.1, 28.9, 26.2, 25.8, 24.8, (10C, adamantane, and octanoyl), 22.6 (C-7), 14.0 ppm (C-8). MS (ESI positive) *m/z* calc. for C_26_H_35_ClO_5_: 462.2 (100%), 463.2 (28%), 464.2 (32%); found: 485.2 (100%) [M + Na]^+^, 486.3 (50%) [M + Na]^+^, 487.3 (55%) [M + Na]^+^, respectively.

Compound **3**

Enol ether **10** (18 mg, 0.034 mmol) and a few milligrams of hematoporphyrin (4 mg, 20% *w*/*w*) were dissolved in 2 mL of THF, reacted for 1.5 h and worked up according to the general procedure. The crude material was purified by column chromatography (hexane:CH_2_Cl_2_:EtOAc, 80:15:2, *v*:*v*:*v*), affording compound **3** (14 mg, 74% yield) as colorless oil. The product was isolated as a mixture of diastereomers. ^1^H NMR (CDCl_3_): *δ* 7.72 (d, *J* = 7.8 Hz, 1H, H-4′), 7.48 (d, *J* = 8.6 Hz, 2H, H-10, and H-14), 7.33 (t app, *J* = 8.1 Hz, 1H, H-5′), 7.11 (d, *J* = 8.6 Hz, 2H, H-11, and H-13), 7.07 (d, *J* = 8.3 Hz, 1H, H-6′), 5.15 (s, 2H, CH_2_O), 3.23 (s, 3H, OCH_3_), 3.02 (br s, 1H, adamantane), 2.56 (t app, *J* = 7.5 Hz, 2H, H-2), 2.36 (d, *J* = 12.7 Hz, 1H, adamantane), 2.04 (br s, 1H, adamantane), 1.92–1.23 (overlapping, 21H, 11 adamantane, and 10 octanoyl), 0.90 ppm (tr, *J* = 6.8 Hz, 3H, H-8). ^13^C NMR (CDCl_3_): *δ* 172.2 (C-1), 154.8 (C-1′), 150.5 (C-9), 133.7, 133.5 (2C, C-12, and C-3′), 128.2 (2C, C-10, and C-14), 127.0 (C-5′), 125.6 (C-4′), 122.7 (C-2′), 121.8 (2C, C-11, and C-13), 115.3 (C-6′), 112.1 (C-1″), 96.3 (C-2”), 70.7 (CH_2_O), 49.6 (OCH_3_), 36.7 (adamantane), 34.4 (C-2), 33.9, 33.5, 32.6, 32.3, 31.7, 31.6, 31.5, 29.1, 28.9, 26.2, 25.9 (11C, adamantane, and octanoyl), 24.9 (C-3), 22.6 (C-7), 14.1 ppm (C-8). MS (ESI positive): *m/z* calc. for C_33_H_41_ClO_6_: 568.3 (100%), 569.3 (36%), 570.3 (32%); found: 569.2 (100%) [M + H]^+^, 570.0 (35%)[M + H]^+^, 571.4 (40%) [M + H]^+^, respectively.

Compound **4**

Enol ether **11** (20 mg, 0.034 mmol) and a few milligrams of hematoporphyrin (4 mg, 20% *w/w*) were dissolved in 2 mL of THF, reacted for 3.5 h and worked up according to the general procedure. The crude material was purified by column chromatography (hexane:EtOAc, 9:2, *v*:*v* + 0.001% of HCOOH), affording compound **4** (11 mg, 52% yield) as colorless oil. The product was isolated as a mixture of diastereomers. ^1^H NMR (MeOD): *δ* 7.94–7.88 (overlapping, 2H, H-1″′, and H-4′ or H-5′), 7.81 (d, *J* = 8.4 Hz, 1H, H-4 or H-5′), 7.51 (d, *J* = 8.5 Hz, 2H, H-10, and H-14), 7.09 (d, *J* = 8.5 Hz, 2H, H-11, and H-13), 6.58 (d, *J* = 16.1 Hz, 1H, H-2″′), 5.00 (br s, 2H, CH_2_O), 3.20 (s, 3H, OCH_3_), 2.98 (br s, 1H, adamantane), 2.58 (t, *J* = 7.4 Hz, 2H, H-2), 2.40 (br d, *J* = 12.5 Hz, 1H, adamantane), 2.01 (br s, 1H, adamantane), 1.90–1.23 (overlapping, 21H, 11 adamantane, and 10 octanoyl), 0.92 ppm (t, *J* = 6.6 Hz, 3H, H-8). MS (ESI positive): *m/z* calc. for C_36_H_43_ClO_8_: 638.3 (100%), 639.3 (39%), 640.3 (32%); found: 661.4 (100%) [M + Na]^+^, 662.4 (50%), 663.4 (60%), respectively. All its chemical–physical properties were superimposable with those reported in the literature [18].

### 3.2. Biochemistry

A time course analysis was performed with compounds **2**, **3**, and **4** to obtain the chemiluminescence emission profile. Experiments were performed in Tris-HCl buffer pH 7.4, 1mM EDTA in the presence (0.2 U/well) or in the absence (no enzyme) of three different lipases, CCL, PFL, and MML, and PLE as control. Varioskan LUX Multimode Microplate Reader (Thermo Fisher Scientific, USA) was used to measure light emission every 2 min for 1.5 h. Assays were performed in total white flat bottom microplates in a total volume/well of 200 µL Tris-HCl, 1mM EDTA (10% DMSO), with 10 μM final concentration of the assessed compounds (**2**, **3**, and **4**). In a typical analysis for the correct amount of the enzyme diluted in reaction buffer contained in a well, the luminescent probe dissolved in DMSO was added, and time-course analysis was immediately started. All experiments were performed in triplicate and repeated at least two times. Results are reported as mean RLUs (relative light units). The means, the standard deviations, and the graphs were calculated and produced by Microsoft Excel. 

## 4. Conclusions

The design, the synthesis, and the full chemical–physical characterization of two unreported phenoxy-1,2-dioxetane luminophores, compounds **2** and **3**, together with the not-commercially-available [18,19] compound **4,** are herein reported. In the design of the new compounds **2** and **3**, the adamantylidene 1,2-dioxetane moiety was selected as luminophore moiety due to the direct luminescence production. Moreover, the elimination of the methyl acrylate substituent at the *ortho* position of the phenoxy-dioxetane present in compound **4** allows us obtaining simpler structures and easier and more effective synthesis. Remarkably, a smart and simple protocol giving the probes **2** and **3** in an easy way and in good yield was set up. The luminescence emission profiles of the probes generated by three different commercial lipases evidence two different behaviors. The chemiluminescent signal obtained from the lipase-catalyzed dioxetane **2** and **3** hydrolysis and subsequent decay is characterized by kinetic behavior, consistent with conventional steady-state Michaelis–Menten kinetics (“glow” kinetics). In this case, the light emitted consists of a more stable but usually less intense signal that can be measured over many hours (glow luminescence), depending on the lipase used. Instead, probe **4** gives a very bright signal for a short time (flash luminescence) for all the lipases, probably due to the presence of the acrylic group. The variety of chemiluminescent 1,2-dioxetane substrates (**2**–**4**) described here enables high versatility, allowing optimization of assay formats with the available instrumentation. The prolonged glow emission kinetics simplifies the measurement of the light signals as shown by compounds **2** and **3**, but for high throughput applications can be disadvantageous due to the waiting time to reach the adequate signal intensity. Although the limit may vary from application to application, a desirable time interval for high throughput applications, typically on the order of seconds, can be achieved with compound **4**. In summary, in this study we showed the applicability of the newly synthesized luminescence probes for measuring the activities of lipases, using three model enzymes.

## Data Availability

Not applicable.

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
