# Peer review of "Small-Molecules as Chemiluminescent Probes to Detect Lipase Activity"

_ijms, 2022, doi:10.3390/ijms23169039_

Round 1

Reviewer 1 Report

The authors report the synthesis of new chemiluminescent probes do detect the lipase activity. I think the manuscript is suitable for publication, despite that I think that can be improved with a more explicit comparison between the new probes and the one already reported in literature for detect lipase.

I think that a clearer scheme could be drawn to indicate the several compounds and their precursors, since the authors report several compounds that are used as precursors of the chemiluminescent probes

In Figure S1 the identification of the axes of the graphic is missing. The caption of the others Figures in Supplementary Material is not presented.

Reviewer 2 Report

Authors in their paper describe the sythesis of highly sensitive substrate for lipase and esterase enzymes. They evaluated luminescence emission profile of substrates developed and the enzyme-substrate specificity, using four different commercial enzymes. Manuscript is quite well written and interesting in light of developing new probes for measuring lipase activity. However, I do not understand why did authors cited references 9 - 11 dealing with antiviral activity of perfluorinated sialic acid glycals against Newcastle disease virus?  
